# Stroke-Prone SHR as Experimental Models for Cardiovascular Disease Risk Reduction in Humans

**DOI:** 10.3390/biomedicines10112974

**Published:** 2022-11-18

**Authors:** Yukio Yamori, Miki Sagara, Hideki Mori, Mari Mori

**Affiliations:** 1Institute for World Health Development, Mukogawa Women’s University, Nishinomiya 663-8143, Japan; 2Department of Health Management, School of Health Study Tokai University, Hiratsuka 259-1292, Japan

**Keywords:** taurine, magnesium, 24 h urine, stroke-none spontaneously hypertensive

## Abstract

Since stroke-prone spontaneously hypertensive rats (SHRSP) develop hypertension and stroke without exception, the prevention or reduction of risk by various nutrients was tested on blood pressure and the mortality caused by stroke and cardiovascular diseases (CVD). In addition to sodium (Na) accelerating hypertension and stroke and potassium (K) counteracting the adverse effect of Na, taurine (Tau), rich in seafood, and magnesium (Mg) contained in soy, nuts, grains, etc., were proven to reduce stroke and CVD and improve survival. Therefore, the Cardiovascular Diseases and Alimentary Comparison Study was started in 1985 to explore the association of biomarkers of diet in 24 h urine (24U) with CVD risks, and about 100 males and 100 females aged 48–56 in each of 50 populations were studied until 1995. Linear regression analysis indicated that the 24U Tau/creatinine and Mg/creatinine ratios were inversely associated with body mass index, systolic and diastolic blood pressure, and total cholesterol. In comparison with six Euro-Western regions, 24U Tau and Mg collected from six regions, respectively, in Japan and the Mediterranean countries were significantly higher and were significantly associated with lower CVD risks. Diets rich in Tau and Mg were concluded to be contributory to the prevention of CVD in SHRSP and humans.

## 1. Introduction

We successfully established stroke-prone SHR (SHRSP, 1974) [1] by successive breeding from a spontaneously hypertensive rat (SHR, 1963) [2], which died of hemorrhagic and/or ischemic stroke. Since they developed stroke spontaneously [3], they were regarded as an appropriate model for research not only on the pathophysiological mechanisms of stroke in general [4], lacunar stroke [5]^,^ cerebral small vessel diseases [6], and subcortical ischemic stroke [7,8], but also the post-stroke treatments [9,10]. Further, since they develop stroke genetically similar to humans, they are a useful model for the potential prediction of stroke through the analysis of genes related to stroke [11,12,13,14,15,16,17]. When stroke can be predicted based on the genes in SHRSP and hopefully in humans, stroke will be prevented by nutrition in humans, as first proven experimentally in SHRSP [3].

Stroke became the leading cause of death in Japan in 1960 after tuberculosis became pharmacologically treated. The nutritional situation in Japan over 60 years ago was supposed to cause stroke among cardiovascular diseases (CVD) because of traditional low protein intake characterized by limited meat supply and owing to habitually low calcium (Ca) and/or magnesium (Mg) intake due to low dairy food consumption. Therefore, we focused on the nutritional prevention of stroke and CVD in the newly established SHRSP by feeding them high-protein diets with Ca and Mg. Since the nutritional intakes were objectively estimated epidemiologically by 24 h urine (24U) analyses of the biomarkers in humans [18], we attempted to show whether experimentally beneficial nutrients for preventing stroke in SHRSP would be epidemiologically associated with cardiovascular risk reduction in humans [19].

Therefore, we proposed to the World Health Organization (WHO) an international cooperative study, “Cardiovascular Diseases and Alimentary Comparison (CARDIAC) Study” [20,21], from our WHO Collaborating Center for Research on Primary Prevention of Cardiovascular Diseases, which was designated by WHO in 1983. In response to our proposal, representative researchers from 61 study sites joined the CARDIAC Study [22].

Since our cooperative studies for over 20 years revealed nutritional biomarkers in worldwide collected 24U samples were significantly associated with CVD risk reduction, we further analyzed these biomarkers in some populations known for their longevity, including Japan (J), which keeps the nearly longest average life expectancy in the world, and Mediterranean (M) countries such as Greece, Sicily of Italy, Spain, and Portugal.

In comparison with Euro-Western countries (EW) (Scotland, Ireland, Sweden, plus Canada, New Zealand, and Australia) where immigrants and descendants from their countries have been living. Among these countries, the popular diets of M countries, rich in polyphenols [23] and related nutrients from fruits, vegetables, olive oil, nuts [24], and fish, have been focused on their basic health effects [25], including cognitive function [26]. However, since no data on M diets compared with EW and J diets has been reported on nutritional biomarkers in 24U samples, we compared them with the CVD risks of M, J, and EW populations in the epidemiological study of this article.

## 2. Materials and Methods

(1) SHRSP had been used for various nutrition experiments [3,19,27,28], and in the present study, SHRSP from 6 groups (Table 1 and Figure 1) were given control and soy diets (CD, SD) with Mg or Ca at the age of 7 weeks, thereafter until their natural death, and autopsied for macroscopical and microscopical pathological observation [19].

(2) A health examination was carried out for males and females according to the protocol of the WHO-coordinated Cardiovascular Diseases and Alimentary Comparison (CARDIAC) Study [20,21], and fasting blood and 24U samples were analyzed after anthropological and blood pressure (BP) measurements [20,21,22,29,30]. About 100 males and 100 females in the age range of 48–56 were randomly invited to the CARDIAC Study health examination after informed consent was obtained from the participants. The study design described in detail [20] was approved at the international committee meeting before starting the CARDIAC Study in 1985.

Informed consent was obtained at the reception of the CARDIAC Study Health Examination from volunteer participants, who were asked to sign the first page of the CARDIAC study questionnaire, and the study was conducted according to the guidelines of the Declaration of Helsinki. Urinary biomarkers such as sodium (Na) for salt intake, potassium (K) for vegetable intake, magnesium (Mg) for grains, nuts, soy, and dietary fiber intake, isoflavones for soy intakes, taurine (Tau) for seafood intakes, urea nitrogen for protein intakes, and creatinine (Cre) for checking the completeness of the collection of 24U samples were analyzed in 50 populations in the world, in total 4211 participants (49.7% females F) in 22 countries worldwide, and also 6 J (864, 53.7% F), 6 M (574, 50.2% F), and 6 EW populations (549, 45.9% F) [18,20,21,22].

Obese subjects were defined as those with body mass index (BMI) ≥ 30 kg/m^2^. Participants with hypertension were defined as those with systolic BP (SBP) ≥ 140 mmHg or diastolic BP (DBP) ≥ 90 mm Hg or those who were receiving anti-hypertensive drug therapy. Hypercholesterolemic subjects were defined as those with serum total cholesterol (TC) ≥ 220 mg/dL. General linear models were used to estimate adjusted mean values of BMI, SBP, DBP, and TC across quintiles of the 24 h urinary Tau/Cre (Mg/Cre) ratio after adjustment for age, sex, and use of anti-hypertensive drugs. To evaluate the association of Tau/Cre (Mg/Cre) ratio with cardiovascular disease risk factors, we estimated adjusted odds ratios for obesity, hypercholesterolemia and hypertension in relation to quintiles of Tau/Cre (Mg/Cre) using logistic regression models, adjusting for age and sex as to hypertension and additionally for anti-hypertensive drugs as to obesity and hypercholesterolemia.

ANOVA was used for the comparisons of 24U biomarkers, BMI, SBP, DBP, and TC among the J, M, and EW diets.

## 3. Results


(1)Experimental Prevention of Stroke in SHRSP.


Since SHRSP developed stroke genetically, they were used to observe the effect of various diets on stroke. For example, SHRSP given 1% salt in drinking water developed severe hypertension and stroke within a short period. However, hypertension was attenuated by increasing K intakes, and even a small reduction of the dietary Na/K ratio significantly improved the survival rate [3,29,30]. The adverse effect of salt was attenuated by alginic acid rich in dietary fibers of the seaweed, which absorbed Na to decrease Na intake via the intestine [27].

The effect of a protein-rich diet was proven in SHRSP fed on a high-fish protein diet with excess salt intake from 1% salt in drinking water. SHRSP fed on low or normal protein diet with excess salt all developed severe hypertension and died from stroke within a shorter period [26]. But the incidence of stroke in SHRSP fed on soy or fish protein-rich diet with excess salt was only 10%. We further analyzed the effect of amino-acids rich in fish and noted that Tau attenuated the development of severe hypertension [28].

Extensive life-long studies on the effect of soy protein with Ca and/or Mg on BP and stroke prevention were designed as shown in Table 1 and Figure 1 in SHRSP, given 1% salt in drinking water. In the present study, soy protein diet (SD) and Mg-fortified control diet (CD + Mg) groups were added to the review of our previous long-term studies [19]. In comparison to SHRSP fed on the control diet (CD) and 1% salt in drinking water, SHRSP fed on a soy protein diet (SP) or a Mg-rich diet (CD + Mg) could survive significantly longer. The average lifespans of these 2 groups (299.6, 305.1 days) were over 200 days longer than the CD group. The effect on lifespan of the 0.6% Mg fortification of a CD diet containing 0.2% Mg (CD + Mg) was similar to the survival of the SD group.

Since the lifespan of 0.9% Ca fortification in CD diets containing 0.7% Ca (CD + Ca) was 166.3 days on average, significantly lower than CD + Mg (305.1 days) or SD (299.6 days). The effect of Mg (0.6%) fortification (CD + Mg) was significantly greater than Ca (0.9%) fortification (CD + Ca).

The average lifespan of salt-loaded SHRSP fed on a soy protein, Mg, and Ca diet was the longest, 417.3 ± 20.7 days.

Their lifespans were significantly longer than salt-loaded SHRSP fed on a control protein diet (88.4 ± days), indicating the intakes of soy protein, Mg, and Ca rich diets were preventive against stroke.


(2)The Association of Urinary Biomarkers with Cardiovascular Risks in the WHO-CARDIAC Study


Since hypertension and stroke were accelerated by Na intake and attenuated by Tau and Mg, 24U samples were collected by the WHO-CARDIAC Study to check the associations of these urinary biomarkers with cardiovascular risks. In addition to the well-known association of 24U Na with BP, stroke mortality rates were significantly positively associated with Na/K ratios [29].

Because of experimental evidence for Tau and Mg attenuating severe hypertension and preventing stroke in SHRSP [28,31], all 24U Tau and Mg data from 50 population samples were divided into five groups, and the adjusted mean values of Tau/Cre and Mg/Cre ratios of the quintiles were inversely associated significantly with BMI, SBP, DBP, and TC in linear regression analyses (*p* < 0.001 for the linear trend and <0.001 for each) [32,33].

The Tau/Cre (Mg/Cre) ratio was significantly inversely associated with obesity, hypercholesterolemia, and hypertension (P for linear trend < 0.001 for the association of the Mg/Cre ratio with obesity, hypercholesterolemia, and hypertension and for the association of the Tau/Cre ratio with obesity and hypercholesterolemia, and <0.05 for the association of the Tau/Cre ratio with hypertension). The odds ratios of obesity, hypercholesterolemia, and hypertension among the subjects in the lowest quintile of Tau/Cre (Mg/Cre) were 2.84 (2.49), 2.20 (2.39), and 1.22 (1.49), compared with the highest quintile (Figure 2 and Figure 3).

These CARDIAC study data indicated that higher intakes of Tau and Mg reduced CVD risk and extended lifespan in humans and suggested that nutritional prevention of stroke for extending lifespan in SHRSP might be applicable to humans. Therefore, CARDIAC study data obtained in J, M, and EW countries were reanalyzed for possible associations with CVD risks.

When Tau/Cre and Mg/Cre of both males and females of J and M were compared with EW, Tau and Mg/Cre in 24U were significantly higher in J and M than in EW (Figure 4). Correspondingly to these 24U data, both systolic and diastolic BP (SBP, DBP*) (*data not shown), TC, and non-HDL cholesterol * were significantly lower in J and M than in EW (Figure 5), although J and M should have significantly higher 24U salt (Figure 6). The significantly higher K excretion in 24 U plus the higher Tau/Cre and Mg/Cre ratios were regarded as the biomarkers contributing to the lower BP in M compared with EW (Figure 6 right).

Despite higher salt intake in J than in EW, the merit of J was the significantly lower BMI compared with M and EW with a higher BMI, which was related to high BP (Figure 7).

One of the factors related to lower BMI in J was supposed to be related to significantly higher 24U isoflavone excretion due to higher intake of soybeans (Figure 7 right), because BMI was inversely associated significantly with 24U isoflavone excretions in CARDIAC Study populations [34].

Since J and M showed higher 24U Tau and Mg/Cre and lower SBP, DBP, and TC, CVD risks were compared between individuals with both Tau/Cre and Mg/Cre ratios equal to or higher than their world average (Tau/Cre ≥ 639.4 mmol/g and Mg/Cre ≥ 82.8 mg/g) and those individuals with both ratios lower than the world average.

BMI, SBP, DBP, TC, and the atherogenic index (AI) calculated from non-HDL/HDL were all significantly lower in the individuals with higher Tau and Mg/Cre ratios than in those with lower ratios of Tau and Mg/Cre, indicating the association of these nutrients with cardiovascular risk reduction in humans (Figure 8).

## 4. Discussion

Since Mg- and Tau-rich diets attenuated the development of hypertension and prevented stroke in SHRSP, 24U of Mg and Tau were examined epidemiologically worldwide in 50 CARDIAC Study populations and were further analyzed in the present study in J and M in comparison with EW populations.

As for Ca, which was proven to prevent stroke and extend lifespan in SHRSP, low Ca intake decreases plasmatic Ca concentration, which stimulates parathyroid hormone (PTH) and renin, angiotensin, and aldosterone secretion to raise BP [35]. Therefore, increased Ca intake attenuates the development of hypertension and stroke. However, since Ca in 24U does not reflect Ca intake in humans [36,37] and is influenced by various factors [38], 24U Ca was not analyzed for its association with the risks of CVD in this study.

Mg and Tau were shown to be related to CVD risks such as hypertension, obesity, and cholesterol-related atherosclerosis.

As for hypertension, Mg activates Na-K ATPase to control electrolyte balance in the cell [39], and therefore the supplementation in the diet reduced intracellular Ca and Na and lowered BP experimentally in SHRSP [31] and clinically in patients with mild hypertension [40]. Tau was observed to decrease BP in SHR and SHRSP [28], and its antihypertensive effect was ascribed to sympathetic modulation [41,42].

In relation to obesity, low Mg status was observed more often in obese individuals [43], and Mg intake was inversely associated with waist size in young Americans [44]. Tau/Cre in 24U was inversely related to obesity in the present study, and the supplementation of Tau decreased body weight in obese mice [45] and clinically in overweight subjects [46].

Mg intake was correlated with the intake of dietary fibers [47], which lowered serum TC [48]. Tau supplementation decreased the effect of a high-fat diet inducing hyperlipidemia in SHRSP and other experimental models [49] by the mechanism of Tau acceleration of bile acid conjugation with cholesterol [50].

Mg and Tau are richly contained in the natural diets, which were obtained commonly from the sea and the mountains even in the Paleolithic period [51]. The recent investigation of a prehistoric kitchen midden in Japan indicated that Mg-rich nuts and seeds, as well as Tau-rich fish and shellfish, were commonly consumed between 5000 and 12,000 years ago [52], and therefore it was speculated that there was no current health problem related to hypertension, obesity, and atherosclerosis. According to the evolutionary concept of human nutrition, such cardiovascular risks as hypertension, obesity, and lipidemia were supposed to be less prevalent in the prehistoric era, and these risks were demonstrated by the present study to be inversely associated with Tau and Mg, the biomarkers of seafood and nuts or seeds, indicating these nutrients commonly taken in the past may potentially reduce current cardiovascular risks. The Japanese are well-known for having the world’s longest average life expectancy, which they have maintained as the top-ranking country for the last 30 years (WHO 2016) [53]. The recent evaluation of Japanese dietary intakes by worldwide urinary biomarker analyses revealed the common consumption of soy isoflavones and seafood Tau was also associated with higher Mg intake [54]. Therefore, a Japanese diet containing commonly Mg and Tau may potentially be related to lower cardiovascular risks, which contribute to their longevity.

In conclusion, Tau and Mg, which were effective for reducing cardiovascular diseases in SHRSPs developing genetically transmitted stroke, were epidemiologically associated with lower CVD risks of obesity, hypertension, and hypercholesterolemia worldwide and were proven in the present study to be the nutritional merits of the diet of the J and M populations, which were known for their relatively longer average life expectancy. However, J and M diets contained more salt; therefore, Tau and Mg rich diets with less salt should be recommended for health promotion with fewer cardiovascular diseases.

## Figures and Tables

**Figure 1 biomedicines-10-02974-f001:**
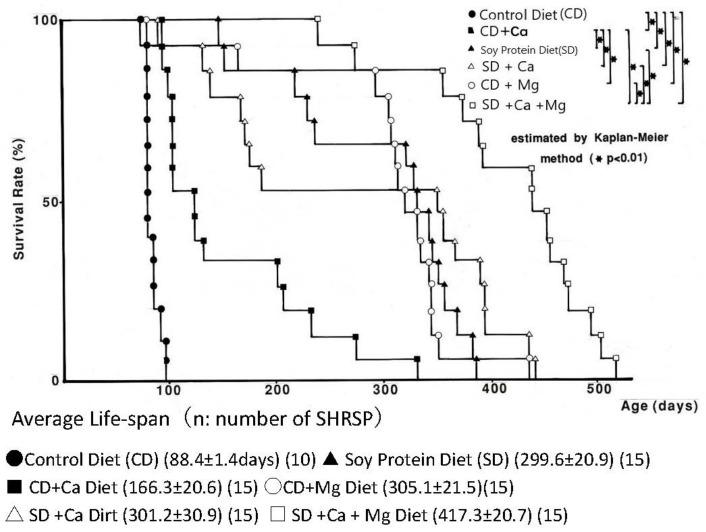
Effect of soy protein, Ca, Mg, and combined diets on the survival rate of salt-loaded SHRSP.

**Figure 2 biomedicines-10-02974-f002:**
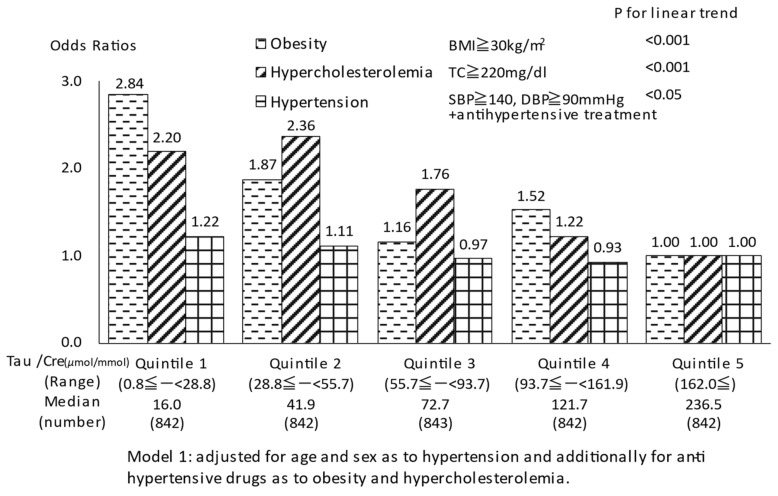
Quintile of Tau/Cre and odds ratios for CVD risks.

**Figure 3 biomedicines-10-02974-f003:**
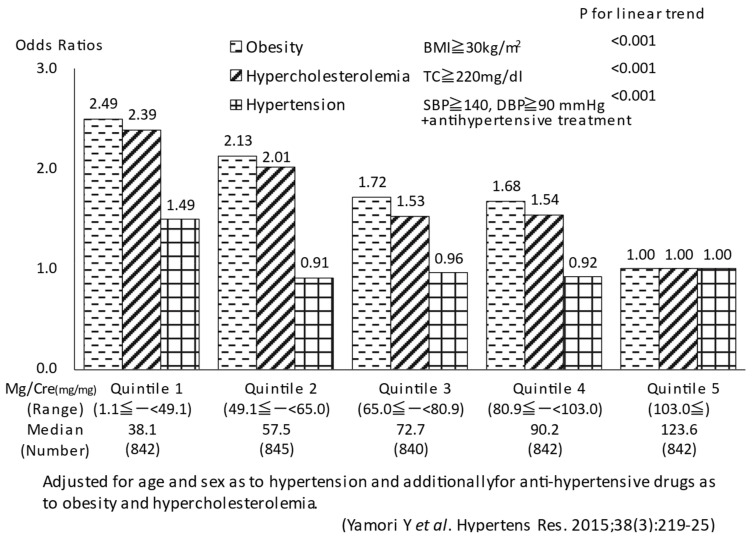
Quintile of Mg/Cre and odds ratios for CVD risks [33].

**Figure 4 biomedicines-10-02974-f004:**
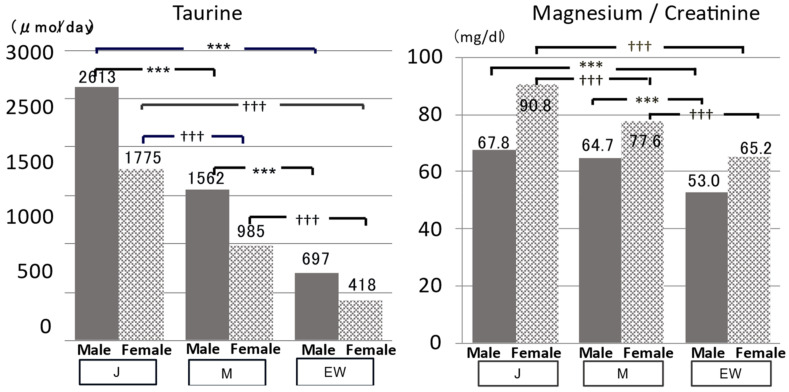
Japanese (J), Mediterranean (M), and Euro-Western (EW) diet populations compared by 24 h urine: Common merits of J and M. J: Aomori, Toyama, Shimane, Chiba, Saga, and Okinawa Prefectures are in Japan. M: Greece, Italy (2), Spain (2), and Portugal. EW: Scotland, Ireland, Sweden, Canada, New Zealand, and Australia. Significant difference: *** *p* < 0.001, ††† *p* < 0.001.

**Figure 5 biomedicines-10-02974-f005:**
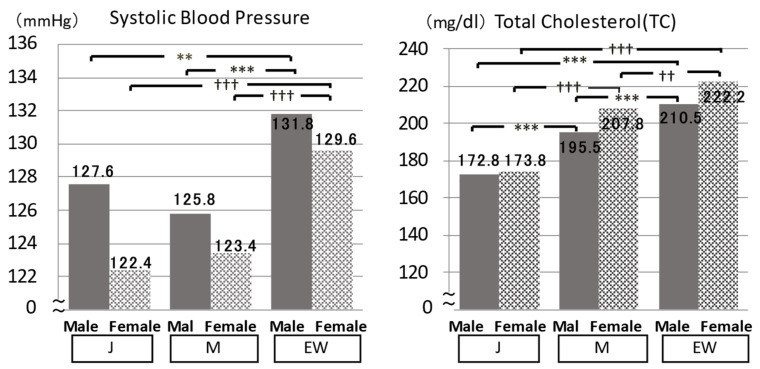
Cardiovascular risks of Japanese (J), Mediterranean (M), and Euro-Western (EW) diet populations: Common merits of J and M compared with EW. Significant difference: ** *p* < 0.01, *** *p* < 0.001, †† *p* < 0.01, ††† *p* < 0.001.

**Figure 6 biomedicines-10-02974-f006:**
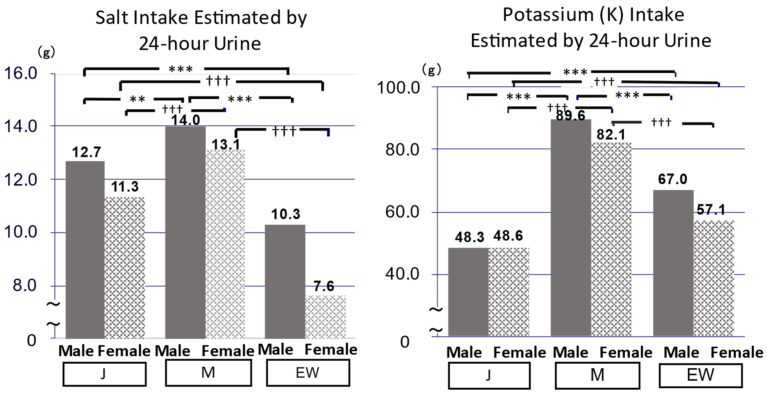
Common demerits of Japanese (J) and Mediterranean (M) diet populations and merits of the Mediterranean (M) diet population (right). Significant difference: ** *p* < 0.01, *** *p* < 0.001, ††† *p* < 0.001.

**Figure 7 biomedicines-10-02974-f007:**
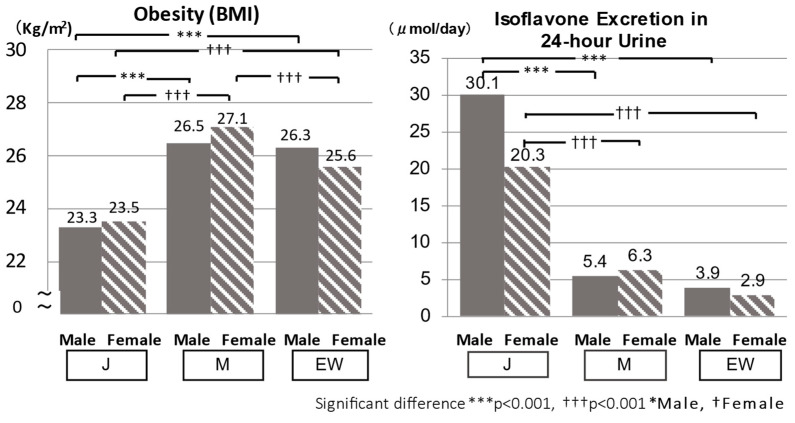
Merit of the Japanese (J) diet population: Low BMI and high isoflavones in 24 h urine excretion in J.

**Figure 8 biomedicines-10-02974-f008:**
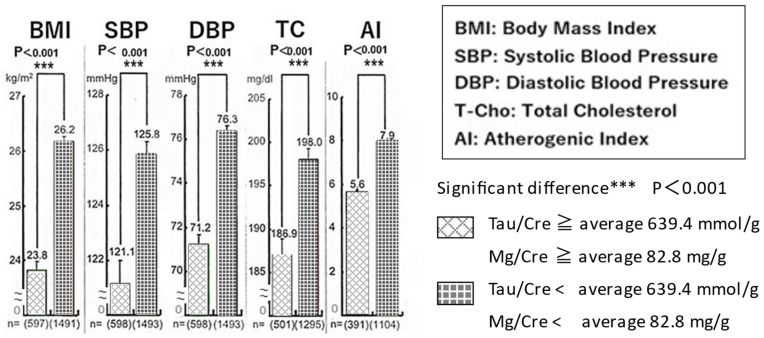
Lower cardiovascular risks were significantly associated with higher Tau (Tau/Cre) and Mg (Mg/Cre) excretions in 24 h urine.

**Table 1 biomedicines-10-02974-t001:** Contents of dietary factors in each group of SHRSP.

Group	N	Dietary Contents	Ca, Mg Contents
Control Diet (CD)	10	Control Diet (Crude Protein: 24.6%)	Ca 0.7%, Mg 0.2%
CD + Ca	15	CD (Ca 0.7%, Mg 0.2%) + Ca (0.9%)	Ca 1.6%, Mg 0.2%
CD + Mg	15	CD (Ca 0.7%, Mg 0.2%) + Mg (0.6%)	Ca 0.7%, Mg 0.8%
Soy Protein Diet (SD)	15	Soy Protein Diet (Soy Protein: 24.6%)	Ca 0.7%, Mg 0.2%
SD + Ca	15	SD (Ca 0.7%, Mg 0.2%) + Ca (0.9%)	Ca 1.6%, Mg 0.2%
SD + Ca + Mg	15	SD (Ca0.7%, Mg 0.2%) + Ca (0.9%) + Mg (0.6%)	Ca 1.6%, Mg 0.8%

## Data Availability

The data that support the findings of this study are available from the corresponding author upon reasonable request.

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
