# Peer review of "Stroke-Prone SHR as Experimental Models for Cardiovascular Disease Risk Reduction in Humans"

_biomedicines, 2022, doi:10.3390/biomedicines10112974_

Round 1
Reviewer 1 Report
The article by Yamori et al. appears to be a reflection on a large body of work spanning six decades, which highlights the importance of the stroke prone SHR model and its value in highlighting nutritional lifestyle changes leading to stroke prevention. There is a lot of jargon and many acronyms that are not adequately explained, for example the abstract needs to state that the SHR is "spontaneously hypertensive rat"!
The manuscript is hard to follow in its present format as it is a mixture of review and original research. It could be improved by making it clearer what are new data and what are reviews or re-analysis of previously published data. Where you are presenting new reserach this surely needs some ethical review component in the methods, likewise the methods requires a summary of the statistical analysis that has been performed for the data that are presented.
There are minor typographical errors throughout.
Author Response
Author’s Reply to Reviewer 1
Thank you very much for your thoughtful review.
- Since SHR is well-known hypertensive rat model established by our group, we supposed readers of this special issue of Biomedicine knew the acronym of spontaneously hypertensive rats. However, we are sorry to use the acronym only first time in the abstract, although the acronym of SHR is explained in the introduction soon after the abstract.
Therefore, we may change the first sentence of the abstract as follows. “Since stroke-prone spontaneously hypertensive rat (SHRSP) develop ーー” according to your advice.
- We developed SHRSP strain by ourselves and we also started 24-hour urinary biomarker analysis for nutritional epidemiological study over 50 years ago when questionaries about dietary intakes were utilized mainly for nutritional epidemiological studies. Therefore, we should mention our background achievements in this manuscript briefly before reporting our originally developed data.
We would like to ask reviewers to understand the important aspect of the present study is that the preventive effect of some nutrients noted by our basic experimental studies in SHRSP models were also proven beneficial 24U biomarkers epidemiologically by WHO-Coordinating CARDIAC Study. The CARDIAC Study that was successfully carried out world-widely in cooperation with many researchers revealed particularly taurine and magnesium were 24U biomarkers associated with lower cardiovascular risks. Therefore, in this study new comparative analysis of these nutrients in Japanese and Mediterranean populations compared with Euro-Western populations newly indicated these nutrients were proven to have common merits for cardiovascular risk reduction in both populations known for their longevity. We differentially presented our previously reported data as the background of the present analyses.
- As I mentioned previously to our Editors “no individual personal name-identified information of volunteers” were used in this article.
Volunteers participated in WHO-CARDIAC Study Health Examination after they signed their mane on the first page of health examination record following “informed consent” at the reception of health examination. This study was conducted by WHO-Collaborating Center for Research on Primary Prevention of Cardiovascular Diseases designated officially by WHO in 1983 and the guideline of Declaration of Helsinki was applied to this study.
- We revised the introduction materials and methods and results of this article extensively and attempted to improve the discussion as far as possible according to both referees advice.

Reviewer 2 Report
The manuscript presented for the review concerns an important and still relevant issue of the need of CVD risk reduction, however, I have some concerns that are as follows:
- Introduction part does not give sufficient background for the study, the aim is not justified and presented
- Materials and methods - this section must be improved, it is not clear whether manuscript presents the review of previously published works of the authors or separate study
Details about part 1 and 2 are missing - even when described previously should be mentioned briefly (unless this is review work?)
- Results again refer to previously published works
- Majority of references are self-citations by the authors
Author Response
Author’s Reply to Reviewer 2
Thank you very much for your thoughtful review.
We are grateful for your comments on the necessity of further improvement of our manuscript.
We tried to improve our manuscript according to your advice.
① As we mentioned to Reviewer 1、the establishment of unique hypertensive rat model developing stroke genetically was our original contribution to medical science.
Therefore, we should include review of our own past works when we attempted to start farther our present original studies on the comparison of 24U biomarkers in J, M and EW countries.
Moreover, as for nutritional epidemiological studies, we started to use 24-hour urinary biomarkers to evaluate the real nutritional intakes of the populations 50 years ago when questionaries were the most common method of nutritional epidemiology.
The new approach could be started because a simple method to collect 2.5% of the voided urine each time was developed by us for the first time in the world. Therefore, please understand we should cite our own original basic and epidemiological studies in this article and we would greatly appreciate referees’ generosity to allow our selfーcitation over 10 %.
② In “Introduction” the aim and justification of our epidemiological study were mentioned.
Since we could prove some 24-hour urinary (24U) biomarkers were associated with cardiovascular diseases (CVD) risks in our world-wide CARDIAC study, we analyzed in the present study the association of 24U nutritional biomarkers of cardiovascular diseases (CVD) risks in Japan, Mediterranean and Euro-Western countries.
③ “Materials and Methods” contained the explanation of the background in our previous experimental and epidemiological studies to show the necessity of the present study.
The data of urinary biomarkers obtained by world-wide CARDIAC study were reanalyzed for the association with CVD risks in J, M, and EW populations.
④ The results of part 1, particularly the data of the effect of Mg and Soy protein in SHRSP were described in more details because they were supposed to be related with the significantly lower ratio of obesity in Japanese populations compared with M and EW countries. Extremely lower rates of severe obesity (BMI>30) in the Japanese are speculated to be associated with the higher isoflavone excretion in the 24-hour urine of the Japanese, although further studies are needed in the future for the confirmation of this association.

Round 2
Reviewer 2 Report
Thank you very much, I am satisfied with the authors' response.